# Amelioration of Hepatic Steatosis by the Androgen Receptor Inhibitor EPI-001 in Mice and Human Hepatic Cells Is Associated with the Inhibition of CYP2E1

**DOI:** 10.3390/ijms232416063

**Published:** 2022-12-16

**Authors:** Shuqin Wang, Xue Li, Weizhe Xu, Jing Gao, Yin Wang, Xiaoyuan Jia, Gongchu Li, Qiuwei Pan, Kan Chen

**Affiliations:** 1College of Life Sciences and Medicine, Zhejiang Sci-Tech University, Hangzhou 310018, China; 2Department of Gastroenterology & Hepatology, Erasmus University Medical Center, 3015 CE Rotterdam, The Netherlands

**Keywords:** hepatic steatosis, CYP2E1, EPI-001, NAFLD

## Abstract

Nonalcoholic fatty liver disease (NAFLD) is recognized as a metabolic disease characterized by hepatic steatosis. Despite the growing burden of NAFLD, approved pharmacological treatment is lacking. As an inhibitor of androgen receptor (AR), EPI-001 is being explored for the treatment of prostate cancer. This study aimed to investigate the potential of EPI-001 for treating NAFLD in free fatty acids (FFAs)-induced human hepatic cells and high-fat-high-sugar (HFHS)-feeding mice. Our results showed that EPI-001 reduced lipid accumulation in hepatic cells and ameliorated hepatic steatosis in mouse livers. Further exploration suggested that the effect of EPI-001 was associated with CYP2E1-mediated reduction of reactive oxygen species (ROS). This provides encouraging evidence for further studies on EPI-001 therapy for NAFLD.

## 1. Introduction

Nonalcoholic fatty liver disease (NAFLD) is characterized as a condition in which steatosis occurs in ≥5% hepatocytes in the absence of alcohol intake [1]. In recent decades, the occurrence of NAFLD has dramatically increased worldwide [2,3,4]. Furthermore, NAFLD has markedly increased in all segments of the population, including children [5]. This disease is often associated with obesity, diabetes, and cardiovascular complications. Thus, the prevention of NAFLD is an important topic for public health. Currently, there are no FDA-approved drugs available for the treatment of NAFLD [6]. 

Numerous studies have reported that oxidative stress (OS) contributes to the pathogenesis of NAFLD [7,8,9,10]. Notably, OS is induced by excessive reactive oxygen species (ROS). Therefore, the generation of ROS is the crucial step in the pathogenesis of NAFLD. It was demonstrated that cytochrome P450 2E1 (CYP2E1) plays a critical role in ROS generation [11,12]. Recently, increased CYP2E1 protein expression has been observed in fatty liver and non-alcoholic steatohepatitis (NASH) in both humans and rodents [13]. Therefore, finding medications that inhibit CYP2E1 overexpression may be a viable option for developing treatment of NAFLD.

As a Bisphenol A diglicycyl ether (BADGE) derivative, EPI-001 has two chiral centers and is a mixture of four stereoisomers. It was identified as an inhibitor of the androgen receptor (AR), which can inhibit the growth of androgen-sensitive prostate cancer cells by binding with the activation function-1(AF-1) region of AR [14,15]. Through covalent binding with the N-terminal domain (NTD), EPI-001 reduced the protein–protein interaction between AR and the co-regulatory protein p300/CBP, which is necessary for AR-mediated transcriptional activation [16]. Although it is currently awaiting clinical assessment, EPI-001 has the potential to be efficacious in preventing the progression of prostate cancer. No toxicity was observed in animals treated with EPI-001 [17]. It was found that the mechanism of action of EPI-001 inhibiting prostate cancer cells was associated with the selective modulation of peroxisome proliferation-activated receptor (PPAR) [18]. PPAR includes three different isotypes (α, β/δ, and γ), which are master regulators of systemic lipid metabolism through a variety of mechanisms. A series of studies have proven the role of PPARs in regulating liver mitochondrial metabolism in the process from NAFLD to hepatocellular carcinoma (HCC) [19,20,21]. 

In this study, we investigated the potential of EPI-001 for treating and preventing NAFLD using human hepatic cell lines and mouse models. We further investigated the possible mechanisms of action.

## 2. Results

### 2.1. EPI-001 Blocks Lipid Accumulation in Human Hepatic Cells 

To explore the effect of EPI-001 on lipid accumulation, human hepatic cell line WRL68 was employed as a model. The cytotoxicity of EPI-001 or free fatty acids mixture (FFAs) was determined by CCK8 assay (Figure 1A,B), and the optimum concentration was selected. Cells were exposed to a 0.6 mM FFAs mixture to induce lipid accumulation, and then 25 μM of EPI-001 or 200 μM of clofibrate were added. The analysis of fluorescence microscopy (Figure 1C) and flow cytometer (FCM) (Figure 1D,E) showed that FFAs incubation effectively induced lipid accumulation in human hepatic cells, while EPI-001 addition blocked cellular lipid accumulation. In addition, we found that EPI-001 was slightly more effective than clofibrate, a widely used lipid-lowering agent (anti-lipidemic). 

### 2.2. EPI-001 Protects against HFHSD-Induced Hepatic Steatosis in Mice 

To investigate the benefits of EPI-001 in the treatment of hepatic steatosis, we established a mouse model of hepatic steatosis by subjecting them to a HFHS diet for 10 weeks. We then injected EP1-001 into mouse intraperitoneally for 8 weeks (Figure 2A). We found that EPI-001 treatment significantly inhibited the increases of body weight (Figure 2B) and liver weight (Figure 2C) in HFHSD mice. HFHS feeding dramatically increased the serum alanine aminotransferase (ALT) level, while EPI-001 treatment, even the low dose, decreased the level of ALT (Figure 2D). However, either HFHSD or EPI-001 had a negligible influence on the serum level of aspartate aminotransferase (AST) (Figure 2D).

In line with the above results, histological examination of mouse liver demonstrated that EPI-001 treatment apparently ameliorated hepatic steatosis (Figure 3A,C), fibrosis (Figure 3A), and decreased NASH activity (Figure 3B). Biochemistry analysis exhibited that EPI-001 significantly reversed the increases of triglyceride (TG) and total cholesterol (TC) in mouse liver (Figure 3D). Serological detection indicated that EP1-001 significantly reduced the levels of TC and high-density lipoprotein cholesterol-C (HDL-C), while slightly reduced TG and low-density lipoprotein cholesterol-C (LDL-C) (Figure 3E,F). (The baseline of LDL and HDL was 1.8–3.5 mmol/L and >1.55 mmol/L, respectively).

### 2.3. EPI-001 Decreases the mRNA Level of Lipid Synthesis-Related Genes 

The imbalance between lipid synthesis and utilization is considered the cause of hepatic lipid accumulation. To further evaluate the role of EPI-001 in lipid accumulation, mRNA expression of the genes related to lipid synthesis was measured in human hepatic cells and mouse liver samples, respectively. Our results indicated that the levels of fatty acid synthase (FAS), sterol regulatory element-binding protein 1c (SREBP-1c), and acetyl-CoA carboxylase 1 (ACC1) were obviously enhanced by FFAs or HFHSD induction, while decreased by EPI-001 treatment (Figure 4A–C and Figure 5A–C). PPARα plays a key role in upregulating a series of genes related to peroxisomal and mitochondrial β-oxidation. The results showed that the mRNA level of PPARα was decreased both in FFAs-induced human hepatic cells and in HFHSD mice livers. However, the level of PPARα was enhanced by EPI-001 treatment (Figure 4D and Figure 5D). OS is caused by excessive ROS, which is produced by CYP2E1 in mitochondria and microstructure. Thus, the level of CYP2E1 was detected both in vivo and in vitro. These results demonstrated that EPI-001 reversed the upregulation of CYP2E1, which was induced by FFAs exposure or HFHSD feeding (Figure 4E and Figure 5E).

EP1-001 has been reported to be efficacious in the treatment of prostate cancer, where AR is thought to play an increasingly important role in tumor progression [15]. It is reasonable to assume that the effect of EPI-001 is related to AR inhibition. Our experiment showed that although EPI-001 reduced the level of AR, neither FFAs nor HFHSD induced an increase in AR (Figure 4F and Figure 5F). This indicated that the effect of EPI-001 on lipid accumulation is independent of AR inhibition.

### 2.4. The role of EPI-001 in Blocking Lipid Accumulation Is Related to CYP2E1 Inhibition and Oxidation Protection

Previous studies identified that CYP2E1 plays an essential role in the development of NAFLD [22]. Thus, the expression of CYP2E1 protein in the liver of HFHSD mice was measured by immunoblotting assays. Our results indicated that the expression of CYP2E1 was promoted by HFHS feeding, while reduced by EP1-001 administration (Figure 6A). Immunohistochemistry (IHC) staining of mice livers showed that CYP2E1 positive areas were mainly concentrated around the central vein of the hepatic lobule in normal conditions. HFHS feeding expanded the positive regions, which bridged with each other. However, the application of EPI-001 alleviated the bridging of these areas (Figure 6B). Since ROS is thought to be generated by CYP2E1, the levels of ROS were detected with FCM in WRL68 cells. The results were consistent with those determinations of immunoblotting and IHC (Figure 6C). These demonstrated that EPI-001 might reduce the level of ROS by inhibiting CYP2E1, and then block lipid accumulation. 

Superoxide dismutase (SOD) can remove lipid peroxides and free radicals, thereby reducing lipid peroxidation. Meanwhile, as an end product of lipid peroxidation, malondialdehyde (MDA) is often adopted as a marker to detect the degree of lipid peroxidation damage. The measurement of SOD and MDA showed that EPI-001 plays the role of oxidation protection in the livers of HFHSD mice (Figure 6D).

## 3. Discussion

As a common chronic liver disease, NAFLD is increasingly recognized as the hepatic manifestation of insulin resistance and the systemic complex known as metabolic syndrome [23,24,25]. To date, no pharmacological medication has received FDA approval for NAFLD [26].

Recently, the role of CYP2E1 in fatty liver has been indicated [27]. CYP2E1 is expressed in hepatic and extrahepatic tissues in mammalians, mainly distributed around the central hepatic vein [28,29,30]. Under normal conditions, CYP2E1 exhibits a low enzymatic activity, which is not involved in vital functions and is usually redundant [31]. Elevated enzymatic activity of CYP2E1 is critically responsible for the development of alcoholic steatohepatitis (ASH) due to the excessive production of ROS during ethanol metabolization [32,33]. In NAFLD, the significance of elevated CYP2E1 expression was also reported despite the lack of alcohol [34]. Increased hepatic metabolic substrates (such as FFAs and ketone bodies) and mitochondrial dysfunction initiated the expression of CYP2E1 [35]. On the other hand, CYP2E1-dependent ROS increase type I collagen synthesis and promote fibrosis, indicating that CYP2E1 was closely associated with hepatic steatosis and fibrosis [36,37]. In addition, CYP2E1 has been proven to be involved in fat synthesis and metabolism by inhibiting PPARα and enhancing SREBP-1c [38]. In line with this research, our results showed that the expression of CYP2E1 was negatively correlated with PPARα, while positively correlated with SREBP-1c. Nevertheless, the exact molecular mechanism of CYP2E1 in NAFLD needs further exploration.

Since CYP2E1 is considered to play a critical role in the development of NAFLD, it is expected to be a novel therapeutic target in NAFLD. Medications that inhibit the over-expression of CYP2E1 may represent a viable option for developing new treatments for NAFLD. Currently, several available medications have shown an impact on mRNA and/or protein expression and enzymatic activity level of CYP2E1 [39]. Some are in clinical use as therapeutics for diseases other than ASH/NAFLD, but none of them has been developed as a specific inhibitor of CYP2E1. Though a series of CYP2E1 pharmacological inhibitors have been designed, no drug candidate has been explored for treating NAFLD [39]. In the present study, we demonstrated that EPI-001 plays a role in anti-oxidative stress via CYP2E1 inhibition. Based on the protection of liver steatosis by EPI-001 in HFHSD mice, we propose that EPI-001 can be considered as a potential therapeutic agent for fatty liver disease. 

Evidence from longitudinal studies suggested that the incidence of NAFLD is higher in males compared to females [40,41,42]. It has been shown that premenopausal women are equally protected from developing NAFLD [43,44], while ovarian senescence is closely associated with severe steatosis and fibrosis in NASH, which may occur in postmenopausal women, and estrogen supplementation may protect from NAFLD development and progression [45]. Interestingly, another study revealed a sex differentiation in hepatic expression of CYP2E1, with higher levels in intact cyclic females at estrus, and lower in males at levels close to these ovariectomized female mice and in cyclic female mice at metestrus. Both progesterone and estrogens increased the hepatic CYP2E1 mRNA expression in ovariectomized mice, whereas tamoxifen, an antiestrogenic agent, markedly repressed the expression of CYP2E1 in the liver of intact female mice [46]. These studies seem contradictory in explaining the association between NAFLD progression and CYP2E1 expression. After all, the development of NAFLD is complicated, and changes of a single molecule are insufficient to explain the pathogenesis. Taken together, the imbalance of hormones should be carefully considered in the clinical treatment of NAFLD. 

In addition, our results showed that EPI-001 significantly decreased the expression of SREBP-1c both in vivo and in vitro. Previous studies have shown that the transcription factor SREBP-1c plays a crucial role in the development of NASH by regulating the synthesis of fatty acid and triglyceride [47,48]. Increased PPAR-α activity has been shown to prevent the expression of SREBP-1c and lead to a decrease of patatin-like phospholipase domain=containing 3 (PNPLA-3) [49], which is closely related to the risks of hepatic steatosis [50]. Therefore, the effects of EPI-001 on hepatic steatosis may be associated with SREBP-1c and PNPLA-3. On the other hand, the global effects of EPI-001 should be noted. We analyzed the epididymal adipose of these mice and found that EPI-001 significantly reversed the increase of epididymal adipose induced by HFHSD (data not shown). Weight loss, potentially caused by reduced food intake and/or increased energy expenditure, may change the upstream signals of the hepatic signaling pathway.

In summary, this study demonstrated the therapeutic effects of EPI-001 for NAFLD, potential through the inhibition of CYP2E1. Although we have provided both in vitro and in vivo evidence showing the role of EPI-001, the direct target of EPI-001 remains unknown. Further studies are required to move the therapeutic development forward, and to better define the mechanisms of action of EPI-001 in preventing the progression of NAFLD.

## 4. Materials and Methods

### 4.1. Cell Culture and CCK8 Assay

The human hepatic cell line WRL68 was purchased from FuHeng Cell Center (Shanghai, China). Cells were cultured in MEM medium (Hyclone Technologies, Logan, UT, USA), supplemented with 10% fetal bovine serum (Hyclone Technologies), 100 units/mL of penicillin, and 100 μg/mL of streptomycin. The cells were grown in a humidified atmosphere of 5% CO_2_ at 37 °C. 

Cells were planted into 96-well plates at the density of 7000 per well and incubated overnight to attach to the bottom, and then treated with serials concentrations of EPI-001 (10, 25, 50, 100, 150, and 200 μM) or FFAs (0.4, 0.6, 0.8, 1, 1.2 and 1.4 mM) for 72 h. Cell viability was measured by using a Cell Counting Kit-8 assay kit (Dojindo Laboratories, Tokyo, Japan) according to the manufacturer’s instructions. An equal volume of PBS containing the same concentration of DMSO was taken as the control in this experiment. Finally, the absorbance was determined with a microplate reader (Thermo Scientific) at the wavelength of 450 nm. The absorbance of untreated controls was taken as 100% survival. 

### 4.2. Analysis of Lipid Accumulation in Human Hepatic Cells

Cells were seeded to a 6-well plate at the density of 1.5 × 10^6^ per well, and then incubated overnight for adherence. Next, the medium was replaced with freshly prepared medium containing 0.6 mM FFAs. For Nile red staining, cells were incubated with FFAs for 24 h. Subsequently, a series of dilutions of chemicals were added. An equal volume of PBS containing the same concentration of DMSO was taken as the control; 48 h later, cells were fixed with 4% paraformaldehyde and dyed with 1 μg/mL of Nile red for 15 min at room temperature. The nuclear was stained with Hoechst 33,358. Images of lipid droplets were acquired by fluorescence microscopy (Olympus (Tokyo, Japan), IX71-22FL/PH). To determine the lipid content, cells were harvested with trypsin, and then the pellets were incubated with 0.75 μg/mL of Nile red dye. The fluorescence of Nile red was measured by flow cytometer (BD Accuri C6) with the excitation/emission wavelength of 525/585 nm. 

### 4.3. Animal, Diets, and Treatment

The care and use of mice conformed to the “Guide for the Care and Use of Laboratory Animals”, and complied with the Laboratory Animal Ethics Committee of Zhejiang Sci-Tech University.

A total of 24 male C57BL/6J mice aged 3–4 weeks and weighing 14–17 g were purchased from Shanghai SLAC Laboratory Animal Co. LTD (Shanghai, China). Mice were housed in cages (2–4 mice/cage) and had free access to diet and drinking water during the whole experiment. All mice were maintained in a pathogen-free environment at 18–23 °C and ~50% humidity, with a 12 h light–dark cycle. After one week’s adaption, mice were divided into two groups randomly by diet. A normal diet (ND) was given to the mice in the ND group (*n* = 6). Mice in the HFHSD group were provided with a high-fat diet and high-sugar drinking water (HFHSD) (*n* = 18). The ND diet includes 10% fat, 20% protein, 70% carbohydrate, and purified drinking water. The HFHSD diet contains 60% fat, 20% protein, 20% carbohydrate, and 45% fructose (*v*/*v*) drinking water. The food was purchased from the Double Lion Experimental Animal Feed Technology Co., Ltd. (Kepong, Selangor, Malaysia) All mice were weighed weekly.

At week 11, HFHSD mice were subdivided into three groups. They were a model group (*n* = 6) that maintained HFHSD feeding; a low dose EPI group (*n* = 6), and high-dose EPI group (*n* = 6) that kept HFHSD feeding and were subjected to EPI-001 (10 mg/kg or 30 mg/kg of EPI-001). The mice in HFHSD + EPI groups were administered intraperitoneally with EPI-001 every three days. Simultaneously, the HFHSD group was given the same volume of saline. After eight weeks of treatment, the mice were sacrificed under deep anesthesia. Their livers were immediately isolated and weighed. A portion of the liver was fixed in 4% paraformaldehyde for histological investigation, and the remaining was snap-frozen with liquid nitrogen. Blood samples of mice were collected from the orbital vein before death, and the serum was prepared by centrifugation, and finally stored at −20 degrees.

### 4.4. Biochemistry Examination

The serum levels of ALT, AST, TG, TC, and HDL-C/LDL-C were determined according to the manufacturer’s instructions by commercial kits from Ningbo Medical system Biotechnology Co. Ltd. (Ningbo, China).

To measure TG and TC contents in mouse liver, about 100 mg of the liver were extracted from the mouse according to the ratio of 1:9 (*w*/*v*) (liver weight: ethanol). Subsequently, the supernatant was measured by TG/TC assay kit following the manufacturer’s instructions. 

### 4.5. Histological Examination of Mouse Liver

The fixed livers were embedded in paraffin wax, and then cut into 4–6 μm sections. Deparaffinized sections were rehydrated and stained with hematoxylin and eosin (H&E) according to standard procedures. The NASH activities score was performed by a pathologist according to the scoring principle of NAS [51]. 

To analyze liver fibrosis, paraffin sections were stained by Sirius red (SR). After conventional dewaxing to water, the sections were dyed with SR dye for 10 min at room temperature. The results were obtained using a digital three-phase camera microscope (BA400 Digital, Motic China Group Co. Ltd., Beijing, China).

Lipid droplets of livers were visualized by Oil Red O (Sigma-Aldrich, Waltham, MA, USA) staining. The fresh livers were snap-frozen with OCT TissueTek (Plano GmbH, Wetzlar, Germany) and cut into 6–8 μm sections. Afterward, a series of processes were followed, including washing, staining, differentiation, and sealing. The average optical density of Oil Red O staining was analyzed by Image-Plus software 2.0 (Media Cybernetics, Inc. Bethesda, MD, USA). 

For IHC, 0.01 M citrate buffer was used for antigen retrieval. The peroxidase was blocked by 3% hydrogen peroxide. Sections were incubated overnight with primary antibody against CYP2E1 (Abcam, 1:100) at 4 °C. The next day, sections were incubated with horseradish peroxidase (HRP)-conjugated secondary antibody (Service bio goat anti-mouse IgG, 1:100) for 1 h at room temperature. Finally, DAB solution and hematoxylin were employed to visualize the staining. Negative control was carried out by omitting the primary antibody.

### 4.6. RNA Isolation and RT-qPCR Analysis

Total RNA was extracted from cells or liver tissues with an RNA isolation kit according to the instructions (Sigma-Aldrich). First-strand cDNA was synthesized from 1 µg of total RNA using Ready-to-go first strand beads (GE Healthcare). RT-qPCR was performed by using the GoTaq Real-Time qPCR mix (Promega). GAPDH was considered as a reference gene to normalize target gene expression. Fold changes were determined by using 2ΔΔCt and normalized to GAPDH. Finally, the fold changes were obtained by converting the logarithmic scale to an exponential scale (2^ΔΔCt^). The primers are listed in Table 1.

### 4.7. Western Blot Analysis

The protein samples of liver tissue were isolated according to the instructions (Sigma-Aldrich). The content of protein was determined by using the BCA method. Proteins (20 μg) were separated by 12% SDS-PAGE and then transferred to a PVDF membrane (Millipore, Bedford, MA). The membrane was blocked using 5% skim milk buffer, and then incubated with primary antibodies against CYP2E1 (1:1000), AR (Abcam, 1:1000), and GAPDH (CST, 1:5000) overnight at 4 °C. Next, the membranes were incubated with secondary antibodies (Abcam, 1:4000) for 1.5 h at room temperature. The membrane was visualized using the ELC detecting kit (Perking Ekmer Inc., Boston, MA, USA) and the Clinx 6000EXP chemiluminescence image system (Clinx, Shanghai, China).

### 4.8. Detection of Oxidative Stress-Related Indicators

To determine ROS, pellets were harvested and incubated with DCFH-DA (Beijing Solarbio Science & Technology Co., Ltd. 1:1000, Beijing, China) for 30 min at 37 °C. The fluorescence intensity was measured by FCM (Agilent Novocyte Advanteon, Santa Clara, CA, USA) at an excitation/emission wavelength of 488/525 nm.

To detect SOD and MDA in mouse liver, tissue homogenate of mouse liver was prepared and then total protein was determined by the BCA method. Prepare samples to be tested according to the instructions of SOD and MDA Kit (Jian Cheng). Finally, the absorbance was measured at 450 nm (for SOD) or 532 (for MDA) with a microplate reader (Thermo Scientific, St. Louis, MO, USA).

### 4.9. Statistical Analysis

Data were shown as mean ± SEM, and statistical analyses were performed with GraphPad Prism software 9.0.0. Statistical evaluation between two groups was carried out by the Mann–Whitney test. For the comparison among three groups, one-way ANOVA followed by Tukey’s multiple comparisons test was used. Differences were considered statistically significant at the *p*-value < 0.05. (* *p* < 0.05, ** *p* < 0.01, *** *p* < 0.001.)

## 5. Conclusions

EPI-001 alleviates fatty accumulation in the liver, which may be related to CYP2E1-mediated reduction of reactive oxygen species.

## Figures and Tables

**Figure 1 ijms-23-16063-f001:**
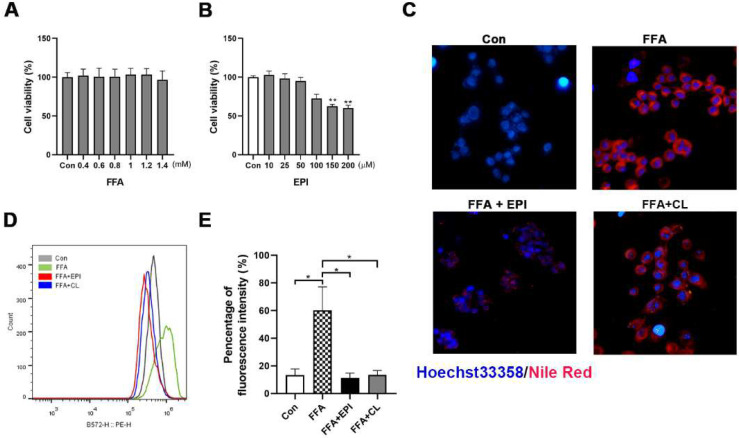
EPI (EPI-001) prevents hepatic lipid accumulation in human hepatic cells. (**A**) CCK8 assay was used to detect the cytotoxicity of free fatty acids (FFAs) in the WRL68 cell line; (**B**) The cytotoxicity of EPI was detected in the WRL68 cell line by using CCK8 assay; (**C**) Representative images of the lipid accumulation (red stained by Nile red) after the treatment of FFAs and EPI-001 (25 μM)/clofibrate (CL) (200 μM) in the WRL68 cell line (400× magnification); (**D**) Nile red fluorescence was detected by flow cytometry (FCM), shown is a representative result from three independent experiments; (**E**) Quantification of FCM. (mean ± SEM, * *p* < 0.05, ** *p* < 0.01). Shown are results from at least three independent experiments.

**Figure 2 ijms-23-16063-f002:**
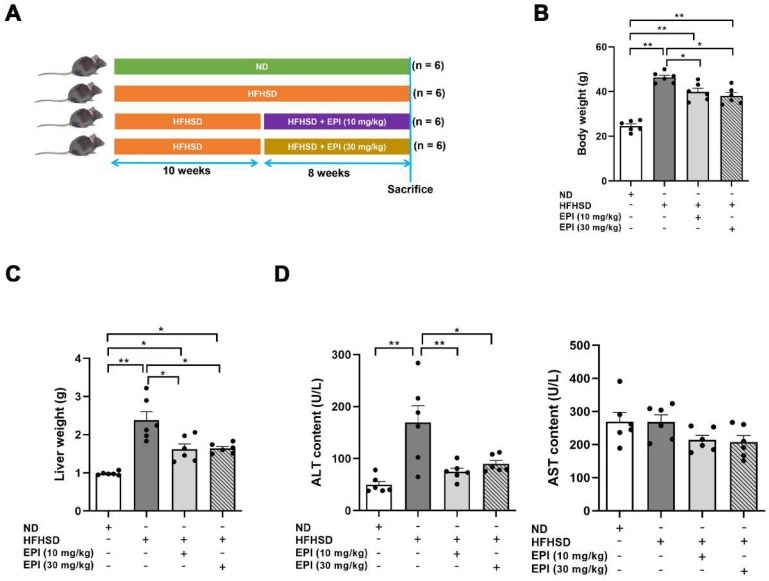
EPI (EPI-001) application ameliorates HFHSD-induced hepatic steatosis. (**A**) Scheme for experiment strategy on a normal diet (ND) or high-fat-high-sugar diet (HFHSD), or HFHSD with EPI treatment (HFHSD + EPI); (**B**) The body weight of mice; (**C**) The wet liver weight of mice; (**D**) The serum levels of ALT and AST in mice (mean ± SEM, * *p* < 0.05, ** *p* < 0.01, *n* = 6).

**Figure 3 ijms-23-16063-f003:**
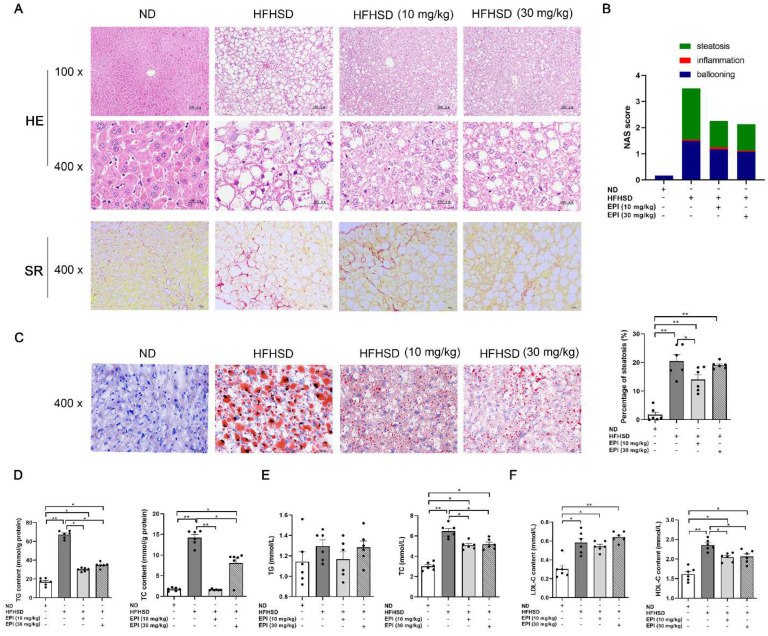
EPI (EPI-001) application ameliorated hepatic steatosis and decreased the levels of biochemical markers in HFHSD-induced mice. (**A**) Representative histologic images of liver sections (HE, hematoxylin and eosin staining, 100× and 400×, scale bar, 100 μm; SR, Sirius red staining, scale bar, 10 μm); (**B**) NAS (NASH activity score) of the HE staining sections; (**C**) Representative images of frozen liver sections stained with Oil Red O (400×, scale bar, 10 μm) and the quantitative results for Oil Red O staining; (**D**) The contents of TG (triglyceride) and TC (total cholesterol) in mouse liver; (**E**) Serum contents of TG and TC in mouse; (**F**) Serum contents of LDL-C (low-density lipoprotein-cholesterol) and HDL-C (high-density lipoprotein cholesterol) in the mouse. (mean ± SEM, * *p* < 0.05, ** *p* < 0.01, *n* = 6 mice per group).

**Figure 4 ijms-23-16063-f004:**
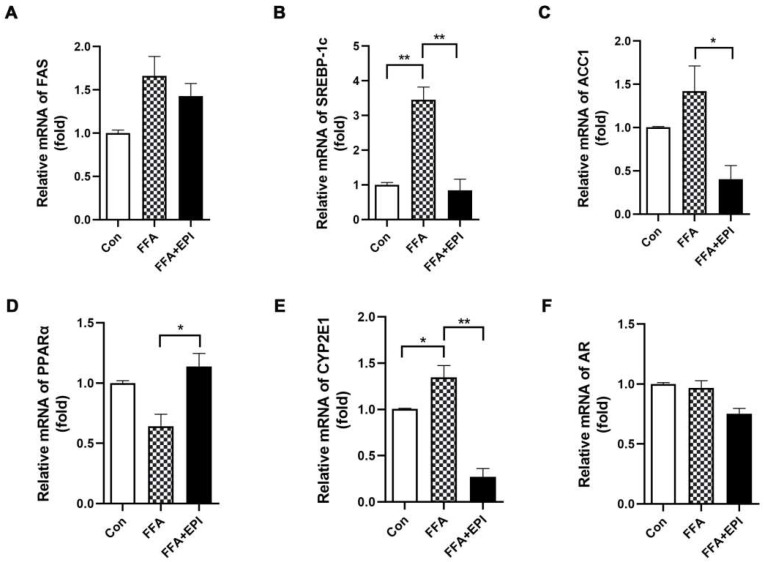
Quantification of lipid synthesis-related gene expression by RT-qPCR assay in human hepatic cells. EPI (EPI-001) inhibited the expression of (**A**) FAS (fatty acid synthase), (**B**) SREBP-1c (sterol regulatory element-binding protein-1c), and (**C**) ACC1 (acetyl-CoA carboxylase 1); (**D**) EPI enhanced the expression of PPARα (peroxisome proliferator-activated receptor-α); (**E**) EPI inhibited the expression of CYP2E1 (Cytochrome P450 2E1); (**F**) The effect of EPI on the expression of AR (androgen receptor). (mean ± SEM, * *p* < 0.05, ** *p* < 0.01, for cell experiment, the result from at least three independent experiments).

**Figure 5 ijms-23-16063-f005:**
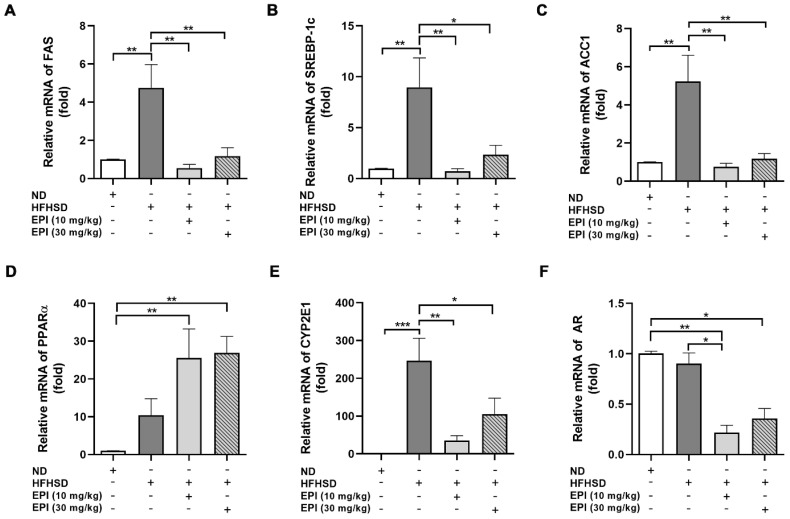
Quantification of lipid synthesis-related gene expression by RT-qPCR assay in mouse liver. EPI (EPI-001) inhibited the expression of (**A**) FAS (fatty acid synthase), (**B**) SREBP-1c (sterol regulatory element-binding protein-1c), and (**C**) ACC1 (acetyl-CoA carboxylase 1); (**D**) EPI enhanced the expression of PPARα (peroxisome proliferator-activated receptor-α); (**E**) EPI inhibited the expression of CYP2E1 (Cytochrome P450 2E1); (**F**) The effect of EPI on the expression of AR (androgen receptor). (mean ± SEM, * *p* < 0.05, ** *p* < 0.01, *** *p* < 0.001, *n* = 6 mice per group).

**Figure 6 ijms-23-16063-f006:**
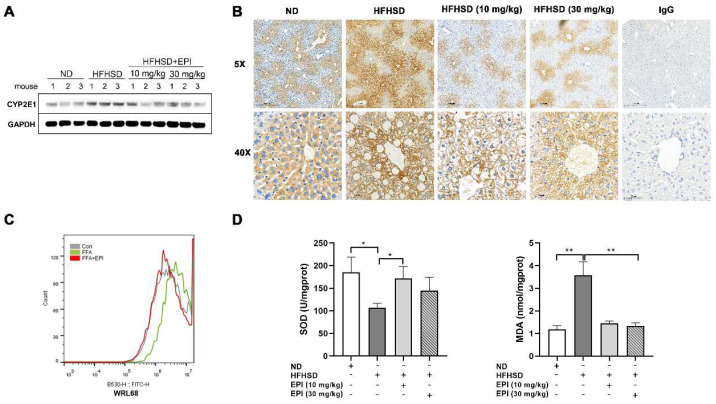
The effects of EPI (EPI-001) ameliorating hepatic steatosis might relate to CYP2E1 (Cytochrome P450 2E1) inhibition. (**A**) Immunoblotting analysis of CYP2E1 in mouse liver; (**B**) Representative immunohistochemical analysis showed that the expression of CYP2E1 was significantly increased in HFHSD mice, while it was decreased by EPI administration; (**C**) Fluorescence intensity of ROS (reactive oxygen species) in human hepatic cells was decreased by FCM; (**D**) The effects of EPI-001 on the contents of SOD (superoxide dismutase) and MDA (malondialdehyde) in mouse liver were detected. (mean ± SEM, * *p* < 0.05, ** *p* < 0.01). Shown are results from at least three independent experiments.

**Table 1 ijms-23-16063-t001:** Primes for RT-qPCR assay.

Gene	Forward Primer	Reverse Primer
FAS Human Mouse	GGACCCAGAATACCAAGTGCAG CTGCGATTCTCCTGGCTGTGAA	GTTGCTGGTGAGTGTGCATTCC CAACAACCATAGGCGATTTCTGG
SREBP-1c Human Mouse	GCGCCTTGACAGGTGAAGTC CGACTACATCCGCTTCTTGCAG	GCCAGGGAAGTCACTGTCTTG CCTCCATAGACACATCTGTGCC
ACC1 Human Mouse	TTCACTCCACCTTGTCAGCGGA GTTCTGTTGGACAACGCCTTCAC	GTCAGAGAAGCAGCCCATCACT GGAGTCACAGAAGCAGCCCATT
PPARα Human Mouse	TCGGCGAGGATAGTTCTGGAAG ACCACTACGGAGTTCACGCATG	GACCACAGGATAAGTCACCGAG GAATCTTGCAGCTCCGATCACAC
CYP2E1 Human Mouse	GAGCACCATCAATCTCTGGACC AGGCTGTCAAGGAGGTGCTACT	CACGGTGATACCGTCCATTGTG AAAACCTCCGCACGTCCTTCCA
AR Human Mouse	ATGGTGAGCAGAGTGCCCTATC TCCAAGACCTATCGAGGAGCG	ATGGTCCCTGGCAGTCTCCAAA GTGGGCTTGAGGAGAACCAT
GAPDH Human Mouse	GTCTCCTCTGACTTCAACAGCG CATCACTGCCACCCAGAAGACTG	ACCACCCTGTTGCTGTAGCCAA ATGCCAGTGAGCTTCCCGTTCAG

## Data Availability

Not applicable.

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
