# Peer review of "Amelioration of Hepatic Steatosis by the Androgen Receptor Inhibitor EPI-001 in Mice and Human Hepatic Cells Is Associated with the Inhibition of CYP2E1"

_ijms, 2022, doi:10.3390/ijms232416063_

Round 1

Reviewer 1 Report

Brief Summary

The study by Wang et al investigated the effects of EPI-001 on ameliorating hepatic steatosis in both an in vitroand an in vivo model. It was shown that EPI-001 treatment reduced lipid content in FFA-treated hepatic cells and liver of HFHS diet-fed mice, and that the effects were associated with CYP2E1 inhibition and reduced oxidative stress.

Overall, the study demonstrated a possible application of an androgen receptor inhibitor EPI-001 on reversing NAFLD and potentially calls for further investigation on its mechanism-of-action (MOA) and therapeutic potential.

General Concept Comments

1.     In Figure 2, the authors show that there is a body and liver weight reduction associated with EPI-001 treatment. This raises a strong possibility of a more global effect of EPI-001. In this case, whole-body scale analysis (e.g., a metabolic cage study) is needed to understand certain important parameters of whole-body metabolism (e.g., food intake, energy expenditure, activity, etc.). Changes in these parameters could be a simple and more direct answer to the effects of EPI-001.

2.     Since liver fat content can be hugely influenced by substrate flux, it would be important to at least measure serum fatty acid and glucose levels. Reduction of these substrates could directly contribute to reduction of hepatic steatosis by reduced substrate delivery even without changes in hepatic signaling pathways. It could also be helpful to look at skeletal muscle and adipose tissue fat content since these tissues have strong crosstalks with the liver and can greatly affect hepatic fat content. In vitro or ex vivomeasurement of liver cell/tissue fatty acid oxidation rates could also be a great addition.

3.     It would greatly increase the significance of the study to add additional support for the causal relationship between the effects of EPI-001 and CYP2E1/ROS reduction. For example, in a cell model, does overexpression of CYP2E1 or induction of ROS abrogates the effects of EPI-001 treatment? In addition, does inhibiting CYP2E1 in hepatic cells replicate the effects of EPI-001 treatment?

Specific Comments

1.     Since a major part of the study is on the CYP2E1 pathway, it would be helpful to add some background on it in the introduction.

2.     In Section 2.1 and Figure 1, it would be important to add concentrations of EPI-001 and clofibrate used in the cell model. These should also be included in the methods section.

3.     The group label of the bar graph in Figure 3C seems mistaken, please check.

4.     In Figure 4F, “mRAN” should be corrected to “mRNA”.

5.     Alternative MOAs should be discussed in the discussion section, since evidence supporting the CYP2E1 pathway isn’t very strong in the study.

Author Response

Point 1: In Figure 2, the authors show that there is a body and liver weight reduction associated with EPI-001 treatment. This raises a strong possibility of a more global effect of EPI-001. In this case, whole-body scale analysis (e.g., a metabolic cage study) is needed to understand certain important parameters of whole-body metabolism (e.g., food intake, energy expenditure, activity, etc.). Changes in these parameters could be a simple and more direct answer to the effects of EPI-001.

Response 1: Thanks for the suggestion, the global effects of EPI are indeed possible. Previous publication reported that EPI exposure reduced BrdU uptake in S-phase cells, with a concomitant increase of cells in G1-phase (PMID: 23722902). Due to limited conditions, we cannot use metabolic cage to detect other parameters. We only analyzed the differences of epididymal adipose and kidney among those groups, and found EPI significantly reversed the increase of epididymal adipose induced by HFHSD, the change in epididymal adipose could be a direct answer to the global effects of EPI-001. However, there was no obvious difference in kidney weight among the 4 groups.

          LFD            HFHSD        HFHSD+EPI (low)  HFHSD+EPI(high)

Weight of epididymal adipose               weight of kidney

Point 2: Since liver fat content can be hugely influenced by substrate flux, it would be important to at least measure serum fatty acid and glucose levels. Reduction of these substrates could directly contribute to reduction of hepatic steatosis by reduced substrate delivery even without changes in hepatic signaling pathways. It could also be helpful to look at skeletal muscle and adipose tissue fat content since these tissues have strong crosstalk’s with the liver and can greatly affect hepatic fat content. In vitro or ex vivo measurement of liver cell/tissue fatty acid oxidation rates could also be a great addition.

Response 2: Thank you for your suggestion. We agree that it is interesting to measure serum fatty and glucose levels to evaluate the effect of EPI. However, C57BL/6 mice were used in this study. The blood volume that could be collected was 600-800 µl per mouse, and only 200-300 µl serum was prepared after final centrifugation. These only enough for basic biochemistry examination. Actually, we also measured the serum levels of cytokines, including IL-6, IL-8 and TNF-α, while the significant changes were not observed (data not shown).

For the analysis of adipose tissue, as shown in the above figures, we observed the epididymal adipose among those groups. The change in epididymal adipose could be direct evidence for the effects of EPI-001.

For the measurement of fatty acid oxidation rate, we also think it is necessary for NAFLD study. PPAR-α plays a vital role in the process of fatty acid β oxidation, our results demonstrated that EPI significantly increased the levels of PPAR-α both in vivo and in vitro. Thus, the association of EPI and PPAR-α mediated fatty acid β oxidation deserves further discussion.

Point 3: It would greatly increase the significance of the study to add additional support for the causal relationship between the effects of EPI-001 and CYP2E1/ROS reduction. For example, in a cell model, does overexpression of CYP2E1 or induction of ROS abrogates the effects of EPI-001 treatment? In addition, does inhibiting CYP2E1 in hepatic cells replicate the effects of EPI-001 treatment?

Response 3: Thanks for the suggestion. We do plan to further explore the effects of EPI-001 on hepatic steatosis in the near future. For example, additional cell lines, such as HepG2 (human), AML-12 (mouse), NCTC1469 (mouse), BRL-3A (rat), and even primary hepatocytes will be used. Simultaneously, the CYP2E1 inducer (such as isoniazid and alcohol) will be used in above hepatic cells.

Point 4: Since a major part of the study is on the CYP2E1 pathway, it would be helpful to add some background on it in the introduction.

Response 4: Thanks for the suggestion, the background of CYP2E1 was added in the introduction section, which was highlighted with yellow color.

Point 5: In Section 2.1 and Figure 1, it would be important to add concentrations of EPI-001 and clofibrate used in the cell model. These should also be included in the methods section.

Response 5: The concentrations of EPI-001 and clofibrate were added in section 2.1 and the legend of fig 1C. These were highlighted with yellow color.

Point 6: The group label of the bar graph in Figure 3C seems mistaken, please check.

Response 6: The group label of figure 3 was corrected as “inflammation”.

Point 7: In Figure 4F, “mRAN” should be corrected to “mRNA”.

Response 7: “mRAN” was corrected in revised manuscript.

Point 8: Alternative MOAs should be discussed in the discussion section, since evidence supporting the CYP2E1 pathway isn’t very strong in the study.

Response 8: The effect of EPI on SREBP1 pathway was added in discussion section, which is highlighted with yellow color.

Reviewer 2 Report

This manuscript demonstrated that EP would be a potential candidate for the management of liver steatosis using both in vitro and in vivo study. However, some points could be addressed.

- The title may be improved to imply what EP is, also the testing system (in vitro, in vivo, in mice, etc) can be expressed.

- Introduction: It was [11]... <<< There is something wrong with the citation.

- Fig1: What is the evidence to conclude that EP was more effective than clofibrate? Because from the graph and flow cytometry, they are non-significant. If the conclusion was based on Nile red staining, the quantification is needed.

- Fig2 and also other figures: Why non-significant (ns) was expressed in the graphs? What was the aim to show them? It made the figure ambiguous.  

- Fig4 and 5: I felt confused to distinguish between results from cell line and mice. It would be better to clearly separate them or indicate them obviously. 

- Fig4: Did the statistical tests were performed? Authors stated in the manuscript that many genes were upregulated; however, they are not statistically significant. So, what are the evidence to conclude those correlations? 

- Fig4: The SEM of control could be shown. Otherwise, it is not necessary to show because the relative value of the control is always 1.

- After reading the results and discussion, I was not convinced to agree that CYP2E1 would be a major factor for the action of EP. How did the author confirm this conclusion? Was it just immunohistochemistry and western blotting using CYP2E1 antibody in mice liver? Also, EP did not downregulate the expession of gene encoding CYP2E1? How about effect on PPAR? 

- Some typos can be found such as inflamation in Fig3.

Author Response

Point 1: The title may be improved to imply what EP is, also the testing system (in vitro, in vivo, in mice, etc) can be expressed.

Response 1: As suggested, the title was revised and highlighted with yellow color.

Point 2: Introduction: It was [11] ... <<< There is something wrong with the citation.

Response 2: The citation was revised and highlighted.

Point 3: Fig1: What is the evidence to conclude that EP was more effective than clofibrate? Because from the graph and flow cytometry, they are non-significant. If the conclusion was based on Nile red staining, the quantification is needed.

Response 3: As shown in fig 1E, we quantified the result of flow cytometry, the bar of EPI is slightly lower than CL, but it is statistically not significant. Now, the sentence is revised as “In addition, we found that EPI-001 was slightly more effective than clofibrate, a widely used lipid-lowering agent (anti-lipidemic).”

Point 4: Fig2 and also other figures: Why non-significant (ns) was expressed in the graphs? What was the aim to show them? It made the figure ambiguous.

Response 4: Thanks for the suggestion. Non-significant (ns) was deleted in the revised graphs.

Point 5: Fig4 and 5: I felt confused to distinguish between results from cell line and mice. It would be better to clearly separate them or indicate them obviously.

Response 5: The previous Fig.4 was separated into Fig.4 (cell line) and Fig.5 (mice). The previous Fig. 5 was changed to Fig 6. The cell experiment in figure 6 (Fig.6C) had been marked with WRL68. The corresponding text of these figures has also been revised.

Point 6: Fig4: Did the statistical tests were performed? Authors stated in the manuscript that many genes were upregulated; however, they are not statistically significant. So, what are the evidence to conclude those correlations?

Response 6: The statistical tests of figure 4 and figure 5 were performed. The changes of SREBP1, ACC1 PPAR-α and CYP2E1 were statistically significantly in cell line, and the changes of FAS, SREBP1, ACC1 PPAR-α, CYP2E1 and AR were statistically significantly in mice.

Point 7: Fig4: The SEM of control could be shown. Otherwise, it is not necessary to show because the relative value of the control is always 1.

Response 7: As suggested, the SEM of control was expressed in figure 4 and figure 5.

Point 8: After reading the results and discussion, I was not convinced to agree that CYP2E1 would be a major factor for the action of EP. How did the author confirm this conclusion? Was it just immunohistochemistry and western blotting using CYP2E1 antibody in mice liver? Also, EP did not downregulate the expression of gene encoding CYP2E1? How about effect on PPAR?

Response 8: We agree that, based on the results of IHC and WB alone, it remains not sufficient to define the exact role of CYP2E1 in the action of EPI-001. We hope to follow-up this research in the near future, for example, studying the expression level of the genes encoding CYP2E1 and testing the effect of CYP2E1 inducer (such as isoniazid and alcohol). It is a good suggestion to investigate that whether EPI may act on PPAR.

Point 9: Some typos can be found such as inflamation in Fig3.

Response 9: The group label of figure 3 B was corrected.

Round 2

Reviewer 1 Report

Brief Summary

The study by Wang et al demonstrated that an androgen receptor inhibitor EPI-001 has an alternative therapeutic potential to treat NAFLD. The effect is associated with a series of signaling pathway changes including the CYP2E1 pathway which regulates ROS production. This calls for further investigation into the hepatic steatosis ameliorating effect and mechanism-of-action (MOA) of EPI-001.

General Concept Comments

1. The authors should add their response 1 on the global effects of EPI-001 to the discussion to expand on the alternative MOAs of EPI-001. Reduced body weight, potentially caused by reduced food intake and/or increased energy expenditure could directly cause liver fat reduction upstream of the hepatic signaling pathway changes.

Specific Comments

1. Line 234 - “through the inhibition of CYP2E1” needs to be toned down since the evidence presented only showed an association between hepatic steatosis reduction and CYP2E1 reduction, so a causal relationship should not be concluded. Simply saying “potentially through the inhibition of CYP2E1” would be better.

Author Response

Point 1:The authors should add their response 1 on the global effects of EPI-001 to the discussion to expand on the alternative MOAs of EPI-001. Reduced body weight, potentially caused by reduced food intake and/or increased energy expenditure could directly cause liver fat reduction upstream of the hepatic signaling pathway changes.

Response 1:Thanks for the suggestion. The global effects of EPI-001 was discussed in the revised manuscript.

Point 2: Line 234 - “through the inhibition of CYP2E1” needs to be toned down since the evidence presented only showed an association between hepatic steatosis reduction and CYP2E1 reduction, so a causal relationship should not be concluded. Simply saying “potentially through the inhibition of CYP2E1” would be better.

Response 2: The sentence was revised and highlighted.
